# Recursive Ectopic Gene Conversion Leads to Elevated DNA Mutation, Gene Loss, and Novel Gene Formation in *Aspergillus*

**DOI:** 10.3390/microorganisms14010033

**Published:** 2025-12-22

**Authors:** Ruojin Wang, Weiwei Liu, Tao Liu, Tianmeng Wang, Huilong Chen, Huilong Qi, Jiangli Wang, Meifang Lan, Xiyin Wang

**Affiliations:** 1School of Public Health and Protective Medicine, North China University of Science and Technology, Tangshan 063210, China; 15100539998@163.com (R.W.); 18093830463@163.com (J.W.); 2College of Mathematics and Science, North China University of Science and Technology, Tangshan 063210, China; 15160251363@163.com (W.L.); liutaocreate@gmail.com (T.L.); 3School of Life Science, North China University of Science and Technology, Tangshan 063210, China; wangtianmeng2023@163.com (T.W.); chenhuilong131@163.com (H.C.); qihuilong_9900@163.com (H.Q.); lanmeifang@ibcas.ac.cn (M.L.)

**Keywords:** gene conversion, paralogous genes, *Aspergillus*, genome instability

## Abstract

Gene conversion contributes to gene copy number changes, DNA mutations, and functional innovation and has been widely reported in three domains of life. However, it has hardly been described in *Aspergillus*, including industrially and commercially important or pathogenic fungi. Here, we revealed multiple sets of homologous genes located in a region of chromosome 1 of *A. flavus*, and its orthologous counterpart of *A. oryzae*. Phylogenetic analysis showed evidence of frequent gene (DNA) conversion between ectopic paralogs in each species, accompanied by prominent point mutations and DNA deletion (from several to hundreds of base pairs). At least two independent cases showed that the converted genes in *A. oryzae* have been repeatedly split into shorter genes by the introduction of stop codons, and then ectopic conversion rendered paralogous genes (regions) to have the same configuration of tandemly located new genes. Inference of nucleotide substitution and ancestral gene content showed that the conversion-affected regions have seen 3.48 times as many substitutions and 4–6 times as many gene losses compared to the non-affected regions. We predicted that a DNA loop between proximal regions, in the common ancestor and inherited by each species, facilitates ectopic gene (DNA) conversion and elevated rates of mutations and losses. Overall, we found that gene conversion proves to be a key factor resulting in genome instability, elevated gene evolutionary rates, and an effective avenue to produce new genes, likely leading to the speciation of two *Aspergillus* lineages.

## 1. Introduction

*Aspergillus flavus* is a common saprophytic fungus, widely found all over the world. It is the main producer of aflatoxins, often contaminating corns, peanuts, tree nuts, cottonseeds, and other important crops before and after harvest [1,2,3]. It is often fatal for individuals with compromised immune function [2,4,5]. However, *A. oryzae*, a member of the same *Aspergillus* group, is widely regarded as safe and commonly used in traditional fermentation industries for the production of condiments such as soy sauce, sake, tofu seasoning, and vinegar [6,7,8]. Like other fungi, these two species were initially distinguished based on their morphological and cultural characteristics [9]. With the development of molecular genetics, there has been a growing interest in their genetic features. The genomes of *A. flavus* and *A. oryzae* have been sequenced, revealing similar genome sizes (36.8 Mb vs. 36.7 Mb) and gene numbers (12,197 vs. 12,079) [10,11,12,13].

Gene duplication is a prevalent phenomenon across the tree of life. In the rice genome, for example, the large number of duplicate genes increases the chances of gene conversion, which can recruit frameshift mutations that shape the evolution of genes and give rise to new genes and functions [14]. In the field of genetics research, scientists have discovered an exciting phenomenon: the emergence of new functionalization may be caused by frameshift mutations [15]. Frameshift mutations occur when nucleotides in the gene sequence are inserted, deleted, or replaced, altering the reading frame of the codons. This results in the generation of a random amino acid sequence until a premature stop codon appears, leading to new functionalities. The emergence of a new gene with a new function from this random sequence is extremely rare, but in terms of genome size, a human NOTCH2 gene copy gained a new function through a 4 bp deletion after the human–chimpanzee split in the process of evolution [16].

Gene conversion is an important mechanism for the recombination and unidirectional transfer of genetic information [17]. Duplicated genes may be subjected to gene conversion, which is a means to transfer genetic information but often keeps two duplicates identical, at least in partial sequences of them. After gene replication, due to the high similarity between the new gene copy and the original gene, unequal cross or gene conversion may occur. Gene conversion involves the unidirectional transfer of genetic information from the “donor” sequence to the highly similar “recipient” sequence, resulting in the “recipient” sequence being replaced in whole or in part by the “donor” sequence, while the “donor” sequence remains unchanged [18]. Gene conversion has been documented in fungi (e.g., yeast) [19], plants (e.g., rice and sorghum) [20], and mammals (e.g., rats and mice) [21]. In eukaryotes, gene conversion is the dominant form of homologous recombination triggered by DNA double-strand breaks (DSBs), by mediating the transfer of genetic information from intact homologous sequences to regions containing DSBs. It can occur within the same chromosome or sister chromosomes (the red and green opsin genes in Old World monkeys) [22], between alleles at the same locus (the MHC gene) [23,24], between homologous non-alleles (*Arabidopsis thaliana*) [25,26], or between ectopic (hetero-chromosomal) genes (Drosophila) proximally located on the same chromosome [27]. Successful gene conversion usually requires a sequence homology of >92%, and typically >95% between the interacting sequences [28]. The estimated lengths of converted sequences vary among organisms. In yeast, converted sequence segments during mitosis (usually >4 kb) are generally larger than those during meiosis (typically 1–2 kb) [29]. In mammals, gene conversion bundles are typically short, with lengths ranging from 200 bp to 1 kb [19].

Although gene conversion has been extensively studied in various organisms, the report on gene conversion in *Aspergillus* has been confined to *A. nidulans,* discussed around 40 years ago [30]. Therefore, here we explored and revealed multiple sets of homologous genes located in a pair of duplicated regions of *A. flavus*, proximally located on a chromosome, and the orthologous counterpart of *A. oryzae*, and cross-genome analysis discovered solid evidence of frequent gene conversion. A closer check of the converted regions showed that gene conversion, characterized by highly similar sequences, is notably accompanied by highly mutated regions in the neighborhood, often involving both point mutations and DNA deletion. This finding is of great significance for studying the origin, differentiation, and adaptability of fungal species, and provides a new perspective for understanding fungal genome evolution.

## 2. Materials and Methods

### 2.1. Materials

Genomic data of *A. flavus* (AF13), *A. oryzae* RIB40 and other *Aspergillus strains* used in this study was downloaded from relevant public databases (Appendix A) [10,11,31,32,33] and preprocessed using Python 3.1.12 scripts for subsequent research.

### 2.2. Sequence Alignment

The alignment of the homologous genes was performed using the MEGA-X (2018) sequence analysis tool [34]. The ClustalW algorithm implemented in MEGA-X was adopted to conduct alignment of the nucleotide sequences of the homologous genes, with parameters set to the software’s default values. The visualization optimization of the alignment results was accomplished using GeneDoc software (GeneDoc 2.7) [35]. This tool was used to format and modify the sequence alignment, clearly marking base matches, mismatches, and deletion sites to enhance the identifiability and presentation clarity of differential features among sequences.

### 2.3. Motif Inference

Based on the genome annotation file (GFF file), the CDSearch function in the Domain and Structure module of TBtools (2024.1.11) [36] was used to predict gene domains, and the Graphics module was further employed to visualize the prediction results, with all parameters set to default values.

### 2.4. Nucleotide Substitution Estimation

IQ-TREE (2.4.0) was used to construct a phylogenetic tree [37]. The nucleotide sequences of AF13, RIB40, and the outgroup genes from *A. fumigatus* were adopted. The Nei–Gojobori method implemented in WGDI (0.5.6) was used to estimate the synonymous nucleotide substitutions (Ks) [38].

### 2.5. Homologous Gene Dotplot

BLAST (2.2.30+) was used to find putative homologs within and between genomes by using genic nucleotide sequences. The E-value threshold of the output results was set to 1 × 10^−5^ to accommodate both duplicated genes and fast-evolving genes, and the output file format (-outfmt) was set to 6 during the specific operation. Based on the previously obtained homologous sequence alignment result file (Blast file), combined with the genome annotation file (GFF file) and chromosome length file (Lens file), we used the—d module implemented in WGDI to draw homologous gene dotplots. The dots of different colors (red, blue, and gray) in the dot matrix represent the level of homology of their gene pairs.

### 2.6. Inference of Gene Conversion

A homologous gene quartet was determined as a pair of paralogous genes within an *Aspergillus* species and their orthologous counterparts, supported by high sequence similarity and corresponding chromosomal locations, in the other species. Then, their putative *A. fumigatus* orthologous gene, being the best hit, was identified as the outgroup of the homologous gene quartet. The protein sequences of each homologous gene quartet and the outgroup were aligned to construct a phylogenetic tree. If there was aberrant observation that the paralogous genes in a species were near one another, but not each near their respective ortholog, we inferred possible gene conversion. Gene conversion was also supported by high sequence similarity between paralogs.

## 3. Results

### 3.1. A Genomic Region Enriched with Homologous Genes

Characterization of gene collinearity between AF13 and *A. oryzae* RIB40 disclosed a shared fragment in AF13 chromosome 1 and *A. oryzae* chromosome 8, which breaks the continuity of gene collinearity between the two chromosomes (Figure 1a). Notably, this fragment was absent from the genomes of the other *Aspergillus* species, NRRL3357 (Figure 1b), *A. niger* CBS513.88, *A. fumigatus* Af293, or *A. nidulans* FGSC A4 (Appendix A).

To find genome structural changes between AF13 and RIB40, a homologous gene dotplot was produced using WGDI, a toolkit to explore gene collinearity in genomes. Gene collinearity analysis characterized the 225 Kb fragment involving 47 genes in AF13 chromosome 1, and the corresponding orthologous region, 192 Kb and 48 genes, in the RIB40 chromosome 8 (Figure 1a).

A closer check of the region showed a distribution of four groups of homologous genes, which form a nested distribution of paralogs, intervened by other genes, in each species (Figure 2a). The first group of homologous genes involves two paralogous AF13 genes in proximal locations, AF131g00222 and AF131g00232 (47.0 Kb apart with nine intervening genes), and their RIB40 orthologs, aor8g00868 and aor8g00878 (46.9 Kb apart with nine intervening genes). Orthology was inferred based on their locations on two chromatic chains (Figure 2b). The paralogous genes, located on different DNA chains, in each species are separated by nine other genes, each of which has respective ortholog in colinear position in the other species.

Neighboring to and encompassing the group 1 paralogs in each species (Figure 1a,b), the second group involves two AF13 paralogous genes in proximity, AF131g00221 and AF131g00233 (52 Kb apart with 11 intervening genes, including the group 1 genes and the 9 intervening ones mentioned above), each, respectively, having homology to two neighboring gene clusters in RIB40, with one cluster containing three genes, aor8g00865, aor8g00866, and aor8g00867 in a region of 3707 bp, and another also containing three genes, aor8g00879, aor8g00880, and aor8g00881, in a region, located on the other DNA chain, of nearly the same length as the former paralogous region (Figure 2c). The two RIB40 paralogous regions are separated by a DNA region nearly the same length as in AF13, containing the same number of intervening orthologous genes.

Two or six genes away from the second group of genes, in a region of 6949bp and 22615bp on one and the other DNA chains of AF13 chromosome 1, the third group contains two paralogous AF13 genes, AF131g00218 and AF131g00240 (96.3 Kb apart with 21 intervening genes including the genes in groups 1 and 2). Group 3 contains three RIB40 homologous genes, aor8g00859, aor8g00860, and aor8g00861, forming a cluster, in a region of 3708 bp (Figure 2d).

Further away from the other three groups of genes in both species, the fourth group contains two paralogous AF13 genes, AF131g00206 and AF131g00246, 174 Kb apart with 39 intervening genes, and their respective RIB40 ortholog, aor8g00853 and aor8g00889, separated by 35 intervening genes in 143 Kb (Figure 2e). The intervening genes contain those from groups 1–3.

As shown above, these four groups of homologs are located in a syntenic and nested manner in AF13 and RIB40 genomes, and therefore show shared gene duplication before the split of the two species. Varied numbers of homologs indicate lineage-specific copy number changes due to gene duplication and gene loss.

Chromosomes 1 and 8 from two species are compared to show the gene collinearity. Red dots between orthologous genes, often forming diagonal lines, display often continuous gene collinearity between chromosomes, and broken points indicate an inserted segment in RIB40 chromosome 8 and AF13 chromosome 1 but not in NRRL3357. Blue dots show secondary homologous gene hits between chromosomes, and gray ones the other hits.

### 3.2. Evidence of Recursive Gene Conversion in Both Species

Phylogenetic analysis and motif characterization support recursive occurrence of gene conversion between paralogous genes in both species. As to the phylogenetic tree constructed for the group 1 genes, with the *A. fumigatus* orthologous gene afu5g7154 as the outgroup, we found that the paralogs from each species are clustered together (Figure 3a). In addition, the paralogs shared higher similarity in sequence and motif composition than the inter-specific homologs. AF131g00232 and AF131g00222 have a sequence similarity of 96.5% and share 16 motifs, while aor8g00868 and aor8g00878 have identical sequences and share 15 motifs. The two AF13 genes have extra Motif10 and Motif12, while the RIB40 genes have extra Motif18 and Motif16. Both sequence similarity and phylogenetic analysis failed to distinguish orthologs between the two species. However, as to the colinear locations on compared chromosomes from two species, from the different transcriptional directions of paralogs in both species, we inferred that AF131g00232 and aor8g00868 are orthologs, and AF131g00222 and aor8g00878 are orthologs. Overall, these facts showed that ectopic gene conversion between paralogs should have occurred in each species, resulting in high similarity in sequence and motif composition between them, while lineage-specific divergence (and conversion) explained the difference between orthologs.

The situation with the group 2 genes also supports the occurrence of gene conversion, but is more complex than that found with group 1 genes (Figure 3b). According to Blast searching, each of the two AF13 genes, AF131g00233 and AF131g00221, on different DNA chains as to their transcriptional directions, seems to have three homologs in RIB40. However, characterization of gene lengths and motif composition displayed that each AF13 gene corresponds to three consecutive RIB40 genes, which proved not to be homologous to one another. Actually, the three RIB40 tandemly arranged genes, aor8g00881, aor8g00880, and aor8g00879, correspond to different but consecutive regions of the AF131g00233, from the upstream to downstream in its transcriptional direction.

Phylogenetic analysis and sequence similarity provided evidence supporting the occurrence of gene conversion. Specifically, for the two RIB40 paralogs, aor8g00881 and aor8g00865, we retrieved their corresponding DNA sequences from the AF13 genes. Using the *A. fumigatus* orthologous gene afu8g9526 as the outgroup, we constructed a phylogenetic tree. Our analysis revealed that paralogous genes (or partial gene sequences) from each species grouped together, which is clear evidence of lineage-specific ectopic gene conversion (Figure 3c). The phylogenetic trees constructed for the other two pairs of paralogous RIB40 genes and their corresponding sequences in AF13 genes, using afu8g9526 as the outgroup, led to the same conclusion of gene conversion. In fact, each of the three RIB40 genes in one region showed nearly 100% sequence similarity to its corresponding paralog in the other region, indicating that gene conversion likely occurred very recently.

A closer check of DNA alignment of each AF13 gene and its corresponding RIB40 genes of group 2 showed that two stop codons were introduced to break the ancestral RIB40 gene into three shorter ones (Figure 4). That is, initially, two duplicated genes existed in the ancestor of AF13 and RIB40, then in RIB40 a duplicated gene was split into three daughter genes due to the introduction of two stop codons, resulting in a tripartite structure. Later on, ectopic gene conversion between paralogous regions homogenized them, rendering the tripartite structure of the split mother gene to its paralogous one (Figure 5).

Notably, in the third group of genes, recursive introduction of stop codons was also found to split a mother RIB40 gene into three short ones (Figure 3d). Actually, there are two AF13 genes, AF131g00218 and AF131g00240, on different DNA chains to their transcriptional directions, while three genes, aor8g00859, aor8g00860, and aor8g00861, are present on the same chain as to AF131g00218. Characterization of gene lengths and motifs suggests that the ancestral RIB40 gene corresponding to AF131g00218 was likely split into three daughter ones. DNA alignment of the AF13 genes and the RIB40 genes showed, las in group 2, that recursive introduction of two stop codons divided the ancestral gene into three daughter genes (Appendix A). A phylogenetic analysis of these genes showed that the AF13 paralogs, with 94.2% similarity, were grouped together, with the RIB40 genes as their outgroups. This inferred that ectopic gene conversion occurred in the AF13 genome. The paralogs corresponding to these three RIB40 genes are presumed to have been lost as a side effect of gene conversion, though gene conversion was not inferred due to the loss of the paralogous genes, as further discussed below (Figure 5).

There was no evidence of ectopic gene conversion in the fourth group of genes. In the phylogenetic tree, with afu1g0070 serving as the outgroup, aor8g00853 clustered with AF131g00246 in one lineage, whereas aor8g00889 and AF131g00206 were placed in a different lineage. This indicates that the paralogs in AF13 or RIB40 did not cluster together, as would be expected if gene conversion had taken place. Motif composition analysis also negated the likelihood of the occurrence of gene conversion. The orthologs, aor8g00853 and AF131g00246, are different in motif order, with Motif16 at the beginning of the former but at the end of latter. For the other two orthologs, AF131g00206 has one more Motif16 than aor8g00889, and Motif5 is positioned after Motif2 in aor8g00889, but after Motif3 in AF131g00206.

For the converted genes inferred above, the DNA regions subjected to conversion were not confined to the exons but also introns, as evidenced by highly similar introns between paralogous genes (Appendix A).

### 3.3. Gene Conversion Incurs Mutations Identifiable Only by Inter-Genomic Comparison

Characterization of sequence variation provides insights into the effects of ectopic gene conversion. We identified identical or highly similar paralogous regions within the same species, but observed significant differences between orthologous regions from two different species (Appendix A).

For group 1, the alignment of sequences revealed about 90% similarity of 1749 bp paralogous regions of the RIB40 genes (aor8g00868 and aor8g00878) and a mere 36.6% in its right-flanking paralogous regions of 41 base pairs including the termination codon. The paralogous regions with lowered similarity were found to be affected by point mutations and indels of 1–3 bp, and further followed by large patches of DNA segment removal in paralogous regions. Significant inter- and intra-genomic variations were identified due to lineage-specific mutations with the group 2 genes. Though the AF13 paralogs (AF131g00222, AF131g00232) were highly similar in the majority of their sequence (97.3% DNA identity), their flanking regions detected differences in 45 base pairs, resulting in a mere 39.1% identity (Chi-square test, *p*-value < 1.0 × 10^−16^). In fact, despite being highly similar between paralogous regions, an AF13-RIB40 comparison revealed that the RIB40 paralogs should have lost three patches of DNA segments of 58 bp, 9 bp, and 18 bp along the transcriptional orientation, and the AF13 paralogs lost a small segment of 4 bp, showing that the AF13 is more stable. In addition, the AF13 sequences were found to extend for another 302 base pairs compared to RIB40 counterparts, showing likely large patches of DNA deletion in the latter. Sequence alignment revealed that an introduction of a stop codon in the RIB40 mother gene seems to have been accompanied by a 4 bp segment of DNA loss before the introduced stop codon, surely resulting from frameshifts caused by DNA deletions, and 302 bp loss after it. For the first pair of RIB40 paralogous genes, aor8g00867 and aor8g00879, the introduced stop codon in each gene was neighbored by highly mutated regions of 140 base pairs, sharing a mere 40% similarity to the corresponding regions of the AF13genes, in contrast to the neighboring highly similar regions (100%, Chi-square test, *p*-value < 1.0 × 10^−16^). The highly mutated regions were affected by extensive point mutations, and two segmental DNA deletions, with six and three base pairs, respectively. In the case of the second pair of paralogous genes, aor8g00866 and aor8g00880, each gene terminates with a short mutated region consisting of six base pairs, which includes the stop codon. Similarly, for the third pair of paralogous genes, aor8g00865 and aor8g00881, each gene terminates with a highly mutated region of 52 base pairs, also encompassing the stop codon. Comparatively, the corresponding region of each AF13 gene has an orthologous identity 36.5%, in contrast to 100% identity in the flanking regions between the RIB40 and AF13 genes (Chi-square test, *p*-value < 1.0 × 10^−16^). After the stop codon of the third RIB40 gene, the corresponding regions of AF13 genes extended for another 700 base pairs, showing a large patch of DNA loss in the RIB40 genes. In addition, it shows that the stop codons of the third RIB40 paralogous genes do not align with those of the AF13 genes, showing that they were also introduced due to DNA mutations, terminating the transcription of the third pair of RIB40 paralogs earlier. In general, these introduced stop codons were each flanked by DNA point mutations and losses, with a length from several base pairs to hundreds of base pairs, and miscellaneous point mutations, suggesting that stop codon introduction was the result of various types and scales of DNA mutations (Figure 5).

These facts with three groups of genes repeatedly showed that although the occurrence of ectopic gene conversion kept highly similar (at least partial) sequences of paralogous genes, it triggered frequent lineage-specific DNA deletion. Notably, these DNA deletions in the converted regions cannot be revealed by intra-genomic comparison, in that its paralogous copy was later often rendered to be identical, though the donor–acceptor relationship could not be revealed. Lineage-specific gene conversion and mutations incurred by it can only be revealed by inter-specific comparison. In fact, divergent orthologous DNA composition showed independent DNA deletions likely accompanied by gene conversion in each lineage.

### 3.4. DNA Loss and Accompanying Gene Conversion Restricts Further Conversion

The recursive gene conversions revealed above between paralogous genes in both species support DNA pairing and recombination between neighboring regions to form a loop structure (Figure 5). Paralogous genes are the basis to form the loop, with the head part to have 9 genes, aor8g00869-aro8g00877 in RIB40, and AF131g00223-AF131g00231 in the AF13. The stem parts, where DNA pairing and conversion occur, should have become shorter and shorter due to DNA losses accompanying recombination. As shown above, gene conversion has occurred between the first two groups of genes in both species. With the third group of paralogous genes, the AF13 genes AF131g00218 and AF131g00240 have been affected by gene conversion, while the corresponding region in RIB40 to AF131g00240 was deleted, likely due to DNA recombination between ectopic regions. Therefore, the loop stem in the latter species was notably shortened.

Though gene conversion could still have occurred between the third group of paralogous genes in AF13, the likelihood should have been much lower, in that DNA similarity has been much reduced between the two arms of the stem from the third to the second groups of genes. In fact, the region on the stem arm contains two genes (AF131g00219-AF131g00220), while that on the other one contains six genes (AF131g00234-AF131g00239).

A much longer stem can be expected to have formed in the ancestral chromosome(s) in both species and their common ancestor. The existence of the fourth group of homologous genes suggests that, although no recent conversion was inferred between the paralogs, ectopic DNA pairing could have occurred between them, or at least the existence of these paralogs in both species showed it to be likely that ectopic homology could have been extending from the first group of paralogs to the fourth group. We reasoned that many genes have been lost during evolution in both species. Not counting species-specific gene splits, we found that there should have been at least 31 genes and 41 genes in the ancestral RIB40 and AF13 regions, respectively. For the inference of ancestral genes, we considered genes in both species, and some of them in one species had no, little, or high similarity to the corresponding DNA region in the other species. The existence of converted paralogous genes intervened by other ones of no, little, or high DNA similarity between affected regions from two species can be explained by likely DNA pairing and conversion accompanied by frequent DNA losses. Thus, we inferred that at least 52 genes should have existed in the corresponding region in the common ancestor, and 19.2% and 40.4% of genes had been lost in AF13 and RIB40, respectively.

### 3.5. Significantly Elevated DNA Mutation as Compared to Other Genomic Regions

The above-revealed conversion-affected regions in both species exhibited significantly elevated Ks values compared to the genome-wide average. We revealed 11397 orthologous genes between two species, and estimated the Ks between each pair of orthologous genes. The Ks values of orthologous genes in the conversion-affected regions were 3.48 times higher than those in the whole-genome scale, showing a significant difference (mean Ks: 0.230 vs. 0.066; T-test, *p*-value = 0.005). At the same time, the conversion-affected regions showed increased gene loss rates. At the whole genome scale, 665 RIB40 and 795 AF13 genes did not find likely orthologs in the other genome, showing a gene loss rate of 5.51% and 6.60%, respectively. Comparatively, the conversion-affected regions have seen 34.6% and 28.5% gene loss in RIB40 and AF13, 6.27 and 4.33 times those at the whole-genome scale, respectively. Here, the estimation of gene loss rates was for those after the split of the species, and considered only the presence of cross-species orthologs. In general, these whole-genome scale analysis supports significantly elevated DNA mutations in the conversion-affected genomic regions in both species.

By revealing proximally located homologous genes in a species, and their orthologs in the other species, named homologous quartets, we tried to locate any other genomic regions subjected to gene conversion. In fact, we revealed 72 homologous quartets exclusive to the above-analyzed conversion-affected regions. These homologous gene quartets were certainly formed by proximal duplicates produced before the split of two species. A check of the topology of the phylogenetic tree reconstructed by a homologous quartet would determine whether there was gene conversion in each species, just like the above analysis. However, no sign of gene conversion was revealed in other regions in two genomes. This showed a scarcity of gene conversion between paralogous genes, justifying the specificity of the above studied regions, which had large patches of paralogous regions containing tens of homologous genes.

## 4. Discussion

As a particular type of genetic recombination, gene conversion plays an important role in the homogenization of gene families [39]. Unlike reciprocal crossing over, gene conversion can change gene frequencies in a population, which is significant for evolution and genetic diversity. Different types of gene conversion have different implications in various biological processes [28].

Here, in the above context, we illustrated that gene conversion occurred at quite high rates between proximal regions in two *Aspergillus* species. In each species, paralogous genes exhibit sequence similarity as high as 100% in the aligned regions, much higher than between the orthologs from the compared species. A high similarity in aligned DNA regions is often accompanied by high point mutation and DNA segmental deletion in regions flanking the converted regions. In fact, we revealed DNA losses in several to tens of base pairs. A further comparison to the orthologous genes in the whole genomes of two species indicated that the genes subjected to gene conversion showed a much higher mutation rate. Undoubtedly, elevated mutation rates act as a driving force in the evolution of the affected genes.

The studied paralogous regions were the hotspot of gene deletion, leading to the genome instability of the genomes [40,41,42]. A comparative analysis of the studied groups of homologous genes found that they may form a loop structure, facilitating ectopic DNA recombination, which incurs DNA conversion and loss. As high as 19.6–40.4% of genes were inferred to have been lost from the affected regions in two species. On a whole-genome scale, the regions affected by gene conversion have experienced 4 to 6 times more gene losses than other genomic regions in the two species. A cross-genomic analysis revealed that these conversion-affected regions have a higher DNA mutation rate, measured by the Ks value, which is 3.48 times higher than that of the non-affected regions. Obviously, the conversion-affected regions have been ever-shrinking due to DNA deletion since their origination in the ancestral species, and have persisted in modern-day organisms. This provides a precious opportunity to observe how a loop structure formed by paralogous DNA of hundreds of base pairs affects genome evolution. These facts show the prominent genomic DNA instability in the local regions, possibly contributing to the speciation of two *Aspergillus* species, with one being an opportunistic pathogen, and the other beneficial to human health.

Gene conversion is an effective way to produce new genes [43]. As shown above, frequent ectopic recombination would introduce stop codons to split an ancestral gene into multiple daughter genes. The introduction of stop codons could result from frameshift mutations or point mutations of non-stop codons, and both avenues are possible for many mutations observed in the above analysis. The new genes produced could be repeated in the paralogous region, as further gene conversion could render the paralogous regions identical [44]. Duplicated new genes can have a significant impact on the regulatory pathways that they belong to [45]. Additionally, if they were to survive, we predict that they may mimic the structure and manner of function of operons observed in bacteria [46]. The peptide(s) encoded by daughter genes may assemble a protein implementing the function of the ancestral gene. This hypothesis could be evaluated in future research.

## Figures and Tables

**Figure 1 microorganisms-14-00033-f001:**
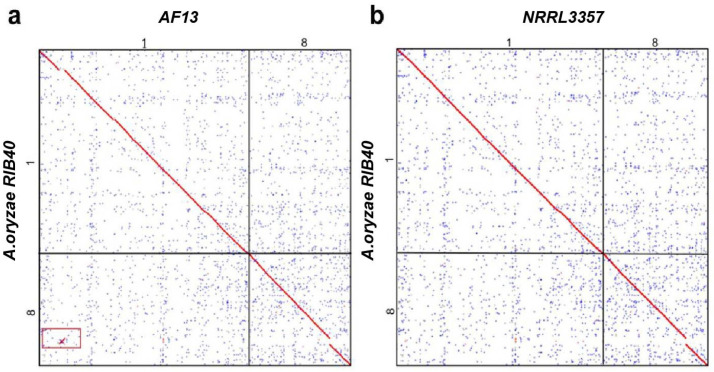
Homologous gene dotplots between strains of *A. flavus* and *A. oryzae*. (**a**) AF13 vs. *A. oryzae* RIB40; (**b**) NRRL3357 vs. *A. oryzae* RIB40.

**Figure 2 microorganisms-14-00033-f002:**
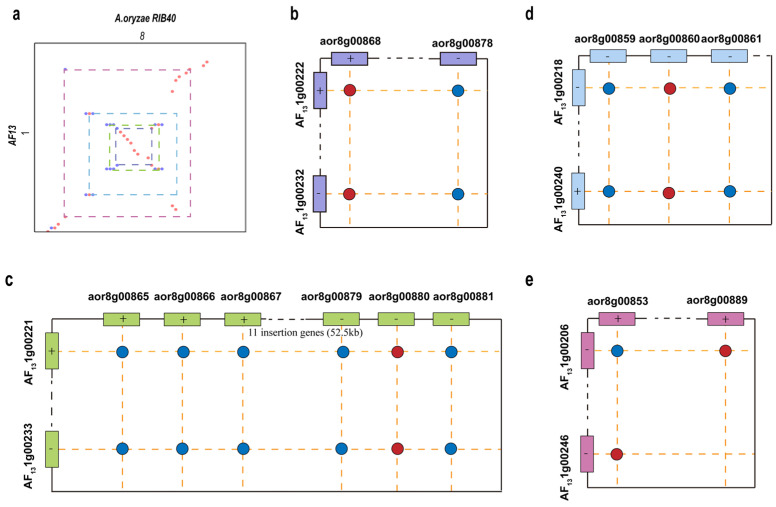
A comparison of duplicated genes in homologous regions from AF13 and RIB40. Red dots represent orthologous genes, while blue dots—generated by the duplication of red dots—represent paralogous genes: (**a**) An insertion of a region arranged on chromosome 8 of RIB40 and chromosome 1 of AF13. Four groups of homologous genes were demonstrated to be nested in the paralogous regions in two genomes. Colored dots show orthologous genes. (**b**) The first group of homologous genes include two genes in AF13, AF131g00222 and AF131g00232, and two in RIB40, aor8g00868 and aor8g00878. A purple box represents Group 1 genes, with a “+” or “−” above the box indicating their distribution on the positive or negative strand. (**c**) The second group of genes include two genes in AF13, AF131g00221 and AF131g00233, and six in RIB40, aor8g00865, aor8g00866, aor8g00867, aor8g00879, aor8g00880, and aor8g00881; a green box represents Group 2 genes, with a “+” or “−” above the box indicating their distribution on the positive or negative strand. (**d**) The third group of genes include two genes in AF13, namely AF131g00218 and AF131g00240, and three genes in RIB40, namely aor8g00859, aor8g00860, and aor8g00861; a blue box represents Group 3 genes, with a “+” or “−” above the box indicating their distribution on the positive or negative strand. (**e**) The fourth gene group includes two genes in AF13, AF131g00206 and AF131g00246, and two in RIB40, aor8g00853 and aor8g00889; a red box represents Group 4 genes, with a “+” or “−” above the box indicating their distribution on the positive or negative strand.

**Figure 3 microorganisms-14-00033-f003:**
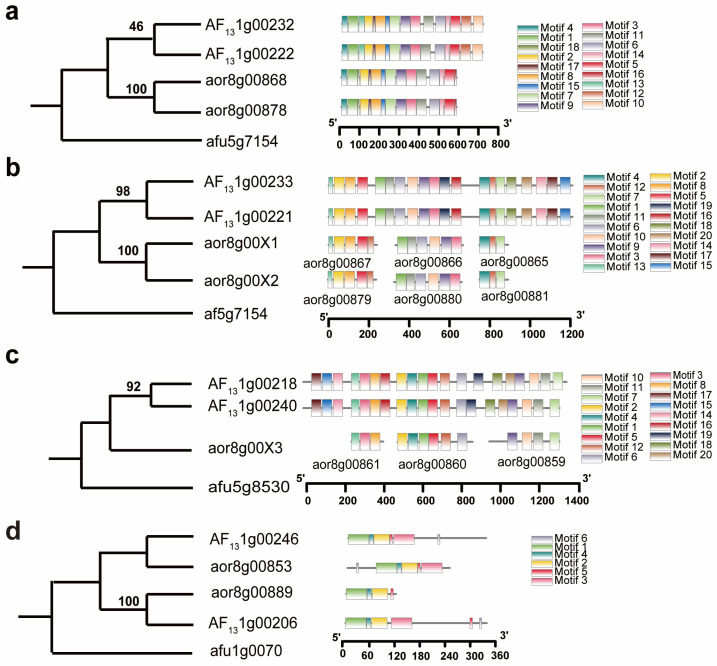
Phylogenetic analysis of homologous genes and inferred motifs. *A. fumigatus* genes are used as the outgroup. (**a**) The homologous gene group 1; (**b**) the homologous gene group 2. Notably, the ancestral gene, here denoted by aor8g00X, in RIB40 were split into three tandem genes, boxed out by a rectangle, due to the introduction of stop codons, and further gene conversion rendered the two paralogous regions identical. (**c**) The homologous gene group 3. Notably, just like the second group, the ancestral gene of the third group, here denoted by aor8g00Y, in RIB40 were split into three tandem genes, boxed out by a rectangle, due to the introduction of stop codons, and further ectopic recombination should have deleted the paralogous region. (**d**) The homologous gene group 4: no trace of gene conversion is inferred.

**Figure 4 microorganisms-14-00033-f004:**
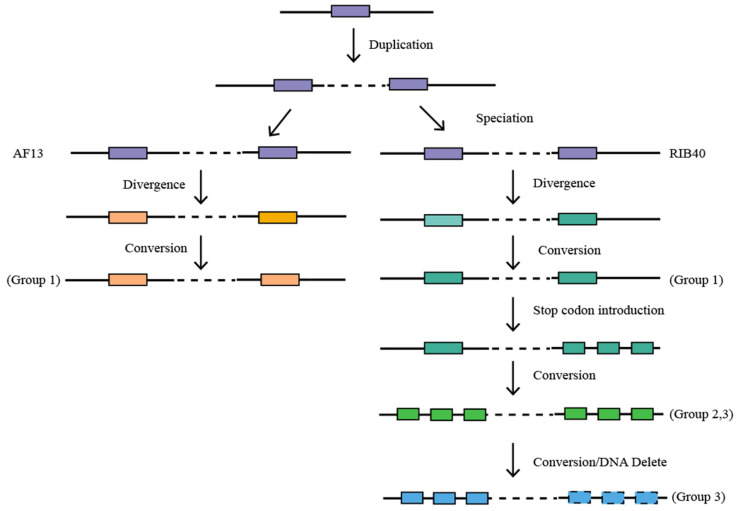
Evolutionary model of recursively converted paralogous genes. Two duplicated genes produced in the common ancestor were inherited by AF13 and RIB40, subjected to gene conversion in both species. In RIB40, one duplicated copy was split into three daughter and tandem genes due to the introduction of stop codon, and further conversion rendered the other duplicated copy to have the same configuration of tandem genes. This explains the finding for homologous gene group 2 and 3. For homologous gene group 3, one set of tandem genes were lost later. Changing colors show sequence divergence.

**Figure 5 microorganisms-14-00033-f005:**
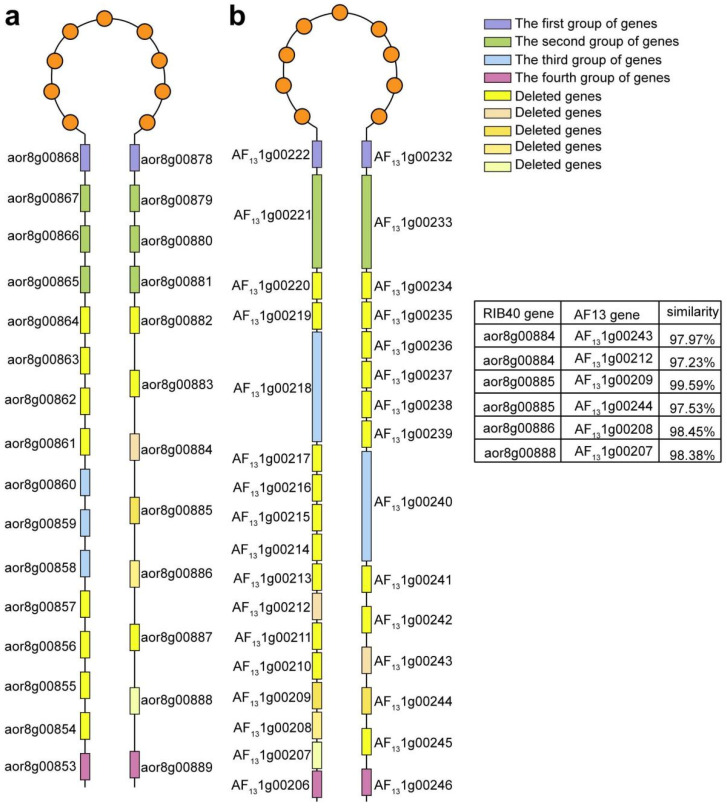
A loop structure formed between proximally duplicated regions in each *Aspergillus* species. Genes are shown as rectangles in the loop stem or as circles in the loop head. Genes in the studied homologous groups or deleted ones are distinctively colored. The similarity between homologous genes from two species, not including the studied four groups, are shown in an embedded table. (**a**) *A. oryzae* (RIB40); (**b**) *A. flavus* (AF13).

## Data Availability

The original contributions presented in the study are included in the article/Appendix A, further inquiries can be directed to the corresponding author.

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
