# Peer review of "Recursive Ectopic Gene Conversion Leads to Elevated DNA Mutation, Gene Loss, and Novel Gene Formation in Aspergillus"

_microorganisms, 2025, doi:10.3390/microorganisms14010033_

Round 1
Reviewer 1 Report
Comments and Suggestions for Authors
The manuscript presents a detailed investigation into the role of ectopic gene conversion in genome instability and gene evolution in Aspergillus flavus and A. oryzae. This is an intriguing and novel topic, as gene conversion in Aspergillus remains underexplored. The work is data-rich, technically competent, and potentially impactful for understanding fungal genome evolution and speciation. The study is compelling and worth publishing, but revisions are needed for clarity, presentation, and better contextual framing of findings. Addressing the above issues will strengthen both readability and impact.
1. The text is dense and sometimes redundant, particularly in the introduction and results. Consider streamlining explanations, avoiding repetitive sentences (e.g., gene conversion mechanisms are explained multiple times with overlapping phrasing).
2. Use consistent naming for species (e.g., A. flavus, AF13) and avoid alternating terms that might confuse readers.
3. Clarify terms such as “homologous quartets” early in the text.
4. Minor grammatical and syntax errors throughout (e.g., “la er” instead of “latter”).
5. The paper provides Chi-square and t-test P-values; ensure appropriate context and explanation for non-specialist readers.
6. Ensure that all figures referenced (e.g., Fig. 1a, Fig. 2d, Fig. 5) are included in the submission.
7. Include proper legends and clarity for supplementary figures cited (e.g., Supplemental Fig. 1 and 2), as they are crucial to support claims about motif and intron conservation.
8. While genome instability and gene novelty are mentioned, the broader functional implications (e.g., in pathogenicity, industrial relevance) could be explored further in the Discussion.
7. The document shows multiple signs of unclean “track changes” or automated editing that weren’t properly finalized. Consider professional copyediting before resubmission.
For example: Terms like “co on-seeds” (should be “cotton-seeds”) show evidence of character deletion.
Also, Incorrect: “Aspergillus. flavus and A. oryzae”; Expected: "Aspergillus flavus and A. oryzae"
Author Response
Comments 1. [The manuscript presents a detailed investigation into the role of ectopic gene conversion in genome instability and gene evolution in Aspergillus flavus and A. oryzae. This is an intriguing and novel topic, as gene conversion in Aspergillus remains underexplored. The work is data-rich, technically competent, and potentially impactful for understanding fungal genome evolution and speciation. The study is compelling and worth publishing, but revisions are needed for clarity, presentation, and better contextual framing of findings. Addressing the above issues will strengthen both readability and impact.]
Response 1: [We sincerely thank you for your positive evaluation and encouraging comments to our manuscript. We greatly appreciate your recognition of the novelty, data richness, and potential impact of our work on ectopic gene conversion in Aspergillus flavus and A. oryzae.]
Comments 2. [The text is dense and sometimes redundant, particularly in the introduction and results. Consider streamlining explanations, avoiding repetitive sentences (e.g., gene conversion mechanisms are explained multiple times with overlapping phrasing).]
Response 2: [Thank you for your constructive feedback regarding the need to condense lengthy sections in the introduction and results. We have revised the manuscript and removed redundant statements in the relevant sections to enhance overall conciseness and readability.]
Comments 3. [Use consistent naming for species (e.g.,A. flavus, AF13) and avoid alternating terms that might confuse readers.]
Response 3: [Thank you for the comment and we have now used consistent naming for species.]
Comments 4.[ Clarify terms such as “homologous quartets” early in the text.]
Response 4: [Thank you so much for pointing out this and we have now defined the homologous gene quartet at its earliest mention: “A homologous gene quartet involves a pair of paralogous genes within a Aspergillus species and their orthologous counterparts, supported by high sequence similarity and corresponding chromosomal locations, in the other species. Then, their putative A. fumigatus orthologous gene, being the best hit, was identified as the outgroup of the homologous gene quartet. The protein sequences of each homologous gene quartet and the outgroup were aligned to construct a phylogenetic tree. If there as aberrant observation that the paralogous genes in a species were near one another, but not each near their respective ortholog, there inferred possible gene conversion”.]
Comments 5. [Minor grammatical and syntax errors throughout (e.g., “la er” instead of “latter”).]
Response 5: [Thank you for pointing out the minor grammatical and syntax errors (e.g., the typo "la er" for "latter") in the manuscript. We sincerely apologize for these oversights, which were caused by our inadequate proofreading in the initial submission.
To address this issue thoroughly, we have conducted a full-scale grammar and syntax check of the entire manuscript.]
Comments 6. [The paper provides Chi-square and t-test P-values; ensure appropriate context and explanation for non-specialist readers.]
Response 6: [Thank you for the advice, we have doubled checked the context to ensure appropriate explanation for readers.]
Comments 7. [Ensure that all figures referenced (e.g., Fig. 1a, Fig. 2d, Fig. 5) are included in the submission.]
Response 7: [Thank you for your reminder, we have carefully checked the manuscript and confirmed that all figures cited in the text having been fully included in the revised submission, with standardized formatting and clear legends for easy reference.]
Comments 8. [Include proper legends and clarity for supplementary figures cited (e.g., Supplemental Fig. 1 and 2), as they are crucial to support claims about motif and intron conservation.]
Response 8: [Thanks for the advice, we have revised the legends of Supplemental Fig. 1 and Supplemental Fig. 2.]
Comments 9 [While genome instability and gene novelty are mentioned, the broader functional implications (e.g., in pathogenicity, industrial relevance) could be explored further in the Discussion.]
Response 9:[Thank you for this insightful suggestion to strengthen the functional implications of our findings in the Discussion section. ]
Comments 10. [The document shows multiple signs of unclean “track changes” or automated editing that weren’t properly finalized. Consider professional copyediting before resubmission.]
Response 10: [Thank you for the reminder, we have double-checked the manuscript to avoid the problem.]
Reviewer 2 Report
Comments and Suggestions for Authors
Ruojin Wang and colleagues present an article on recursive gene conversion based on in silico analysis and genome comparisons. The article may contain useful information for people working on this field and beyond, however, the hypothesis cannot be fully supported given the lack of wet experiments. Therefore, the emphasis on many conclusions should be lowered.
Although comprehensibly written, still parts of the results are quite difficult to follow, jumping from one figure to the other. Some restructuring of the text and the supporting figures would be welcome. Some more time must be invested in the overall presentation of the figures in order to make them more conclusive and reader friendly.
In A. nidulans, several older articles refer to gene conversion etc., so why the authors did chose to work with A. oryzae or A.flavus? Is anything known on the function of all these genes studied? Without this information I see no point to talk about non- or neo-functionalization etc. Are there any examples of recursive gene conversion in genes of known function?
Please check the spelling of the word Aspergillus throughout the text.
Lines 160-162 are just advising the authors and do not belong to the actual manuscript.
Author Response
Comments 1: [Ruojin Wang and colleagues present an article on recursive gene conversion based on in silico analysis and genome comparisons. The article may contain useful information for people working on this field and beyond, however, the hypothesis cannot be fully supported given the lack of wet experiments. Therefore, the emphasis on many conclusions should be lowered.]
Response 1:[Thank you for your comment. In the revised manuscript, we have double-checked the conclusion and lowered tone. We are going to perform experiments in future research to explore how gene conversion affected the structure of DNA sequences, point mutations and InDels, and try to reveal the detailed mechanisms behind the phenomenon. ]
Comments 2: [Although comprehensibly written, still parts of the results are quite difficult to follow, jumping from one figure to the other. Some restructuring of the text and the supporting figures would be welcome. Some more time must be invested in the overall presentation of the figures in order to make them more conclusive and reader friendly.]
Response 2: [Thank you for your constructive comment on the presentation of results and figures. We have restructured the Results section to align with the figure sequence, and optimized all figures for better clarity and readability.]
Comments 3: [In A. nidulans, several older articles refer to gene conversion etc., so why the authors did chose to work with A. oryzae or A.flavus? Is anything known on the function of all these genes studied? Without this information I see no point to talk about non- or neo-functionalization etc. Are there any examples of recursive gene conversion in genes of known function?Please check the spelling of the word Aspergillus throughout the text.Lines 160-162 are just advising the authors and do not belong to the actual manuscript.]
Response 3: [Thank you for the comment. Actually, we have been studying the pathogenicity of A. flavus to human. As a foundational research, we performed some exploration of A. flavus genome sequence, and compared it to that of A. oryzae and other several relative species. We noted that a large insert fragment exists between A.flavus AF13 and A.oryzae RIB40. This incurred our interest to find whether there is gene conversion between the paralogous genes in each species, which followed gene conversion exploration in our research group, mainly in plants, for about 20 years.
We agree that the inclusion of this information enhances the comprehensiveness of the discussion on fungal genome stability in microbial systems within our study. Below is the revised content supplemented with gene conversion in Aspergillus nidulans(A. nidulans), which has been integrated into Page 2 of the Introduction section of the manuscript.
Thank you for your valuable comment. We fully agree that examples of recursive gene conversion in genes of known function are critical for discussions on non/neo-functionalization. To date, no such explicit cases have been identified in our study, though gene conversion is well-documented in functionally annotated genes. Accordingly, we have revised the relevant wording in the manuscript to accurately reflect this research status.
Our investigation of this fragment revealed a spectrum of genetic variations, including gene conversion, gene loss, and point mutations. Notably, despite their highly similar genome size and architecture, A. oryzae is widely exploited as an industrial fermentation strain, whereas A. flavus is recognized as a toxigenic pathogen. This striking contrast between the two closely related species, coupled with the genetic diversity harbored in the insert fragment, constitutes a key rationale for our research.]
Reviewer 3 Report
Comments and Suggestions for Authors
Dear Authors
The manuscript requires substantial revision before it can be considered for publication. The introduction does not sufficiently establish the scientific context or clearly articulate the research gap, as several key and recent studies are missing. The methodological descriptions are incomplete, limiting the reproducibility of the work, and the presentation of results lacks clarity, organization, and adequate figure/table quality. Several interpretations and conclusions extend beyond what is directly supported by the data and should be tempered and more tightly aligned with the results and existing literature. In addition, the overall readability is hindered by frequent grammatical issues and awkward phrasing, and the manuscript would benefit greatly from comprehensive professional English editing.
minor
-
Redundant sentences: In the Introduction, content describing Aspergillus flavus contamination and pathogenicity appears twice. Compare Lines 21–25 with Lines 26–34, which repeat nearly identical information regarding aflatoxin production and human/animal health impacts.
-
Corrupted character “Ä´” appears multiple times.: At Line 40, “coÄ´on-seeds” should be corrected to “cotton seeds.” Similar corrupted characters appear elsewhere and should be standardized.
-
Duplicated microarray sentence within the same paragraph: The sentence beginning with “In recent studies, the microarray method was used…” appears twice at Lines 21–25 and again at Lines 52–55. Remove one instance to avoid redundancy.
-
Grammar and repetition issues in frameshift mutation description: At Lines 68–78, the explanation contains repetitive wording (“changes to the nucleotides in the gene sequence” repeated) and awkward phrasing. This section should be streamlined for clarity.
-
Typographical error “DSBS” should be corrected : At Line 93, “DSBS” should be corrected to “DSBs” (double-strand breaks) to match standard terminology.
-
Unclear or incorrect grammar in Discussion section: At Lines 472–475, the sentence “The introduction of stop codons could be resulted to frameshift…” is ungrammatical. A correct construction would be “could result from frameshift mutations or point mutations”.
-
Spelling error “mimick.”: At Line 479, the word “mimick” should be corrected to “mimic”.
-
Spacing inconsistencies: Across multiple lines, spacing before citations (e.g., “[14–16]”), hyphenation, and species italicization (Aspergillus flavus, A. oryzae) are inconsistent and should be standardized.
-
Figure and table captions need clearer descriptions: Several captions lack sufficient detail to allow figures/tables to stand alone. Please expand captions to describe datasets, abbreviations, and key results.
-
Minor formatting issues in references: Some references (e.g., Lines 568–575) include inconsistent punctuation or spacing around years, volumes, and page numbers. These should align with the journal’s citation style.
Author Response
Comments 1: [The manuscript requires substantial revision before it can be considered for publication. The introduction does not sufficiently establish the scientific context or clearly articulate the research gap, as several key and recent studies are missing.]
Response 1: [Thank you so much for pointing out this, we have now double polished the Introduction and other sections to ensure fluent description.]
Comments 2: [The methodological descriptions are incomplete, limiting the reproducibility of the work, and the presentation of results lacks clarity, organization, and adequate figure/table quality.]
Response 2: [Thank you for the advice, we have now revised the description of methods, and results, and hope the present manuscript is good in clarity.
Especially, we have now defined the homologous gene quartet at its earliest mention: “A homologous gene quartet involves a pair of paralogous genes within a Aspergillus species and their orthologous counterparts, supported by high sequence similarity and corresponding chromosomal locations, in the other species. Then, their putative A. fumigatus orthologous gene, being the best hit, was identified as the outgroup of the homologous gene quartet. The protein sequences of each homologous gene quartet and the outgroup were aligned to construct a phylogenetic tree. If there as aberrant observation that the paralogous genes in a species were near one another, but not each near their respective ortholog, there inferred possible gene conversion”.
Besides, we have now provided a detailed description of Figure 1. “Figure 1. Homologous gene dotplots between strains of A. flavus and A. oryzae. a: AF13 vs. A. oryzae RIB40; b: NRRL3357 vs. A. oryzae RIB40. Chromosomes 1 and 8 from two species are compared to show the gene collinearity. Red dots between orthologous genes often forming diagonal lines display often continuous gene collinearity between chromosomes, and broken points indicate an inserted segment in RIB40 chromosome 8 and AF13 chromosome 1 but not in NRRL3357. Blue dots show secondary homologous gene hits between chromosomes, and gray ones the other hits”.
We have also polished the other parts of the manuscript, especially the Introduction and Discussion.]
Comments 3: [Several interpretations and conclusions extend beyond what is directly supported by the data and should be tempered and more tightly aligned with the results and existing literature. In addition, the overall readability is hindered by frequent grammatical issues and awkward phrasing, and the manuscript would benefit greatly from comprehensive professional English editing minor].
Response 3:[Thank you very much for carefully reviewing our manuscript and providing valuable constructive feedback. We highly value all the issues you have pointed out. In response to your comments, we have made substantial improvements to the manuscript.]
Comments 4: [Redundant sentences: In the Introduction, content describing Aspergillus flavus contamination and pathogenicity appears twice. Compare Lines 21–25 with Lines 26–34, which repeat nearly identical information regarding aflatoxin production and human/animal health impacts.]
Response 4: [Thanks for the reminder, we have now revised the manuscript.]
Comments 5: [Corrupted character “Ä´” appears multiple times.: At Line 40, “coÄ´on-seeds” should be corrected to “cotton seeds.” Similar corrupted characters appear elsewhere and should be standardized.]
Response 5:[Thank you for the valuable comments, we have revised the manuscript accordingly. ]
Comments 6 :[Duplicated microarray sentence within the same paragraph: The sentence beginning with “In recent studies, the microarray method was used…” appears twice at Lines 21–25 and again at Lines 52–55. Remove one instance to avoid redundancy.]
Response 6:[Thank you for your reminder, we have made revisions accordingly.]
Comments 7: [Grammar and repetition issues in frameshift mutation description: At Lines 68–78, the explanation contains repetitive wording (“changes to the nucleotides in the gene sequence” repeated) and awkward phrasing. This section should be streamlined for clarity.]
Response 7: [Thank you for the advice, we have revised the text to eliminate repetitive phrasing and improve clarity.]
Comments 8 :[Typographical error “DSBS” should be corrected: At Line 93, “DSBS” should be corrected to “DSBs” (double-strand breaks) to match standard terminology.]
Response 8:[Thank you for your reminder, we have made revisions accordingly.]
Comments 9:[ Unclear or incorrect grammar in Discussion section: At Lines 472–475, the sentence “The introduction of stop codons could be resulted to frameshift…” is ungrammatical. A correct construction would be “could result from frameshift mutations or point mutations”.]
Response 9: [Thank you for your reminder, we have made revisions accordingly.]
Comments 10:[ Spelling error “mimick.”: At Line 479, the word “mimick” should be corrected to “mimic”.]
Response 10:[ Thank you for your reminder, we have made revisions accordingly.]
Comments 11: [Spacing inconsistencies: Across multiple lines, spacing before citations (e.g., “[14–16]”), hyphenation, and species italicization (Aspergillus flavus, A. oryzae) are inconsistent and should be standardized.]
Response 11:[Thank you for your reminder, we have made revisions accordingly.]
Comments 12:[ Figure and table captions need clearer descriptions: Several captions lack sufficient detail to allow figures/tables to stand alone. Please expand captions to describe datasets, abbreviations, and key results.]
Response 12:[ Thanks so much, we have revised all the figure and table captions in light of your comments. Specifically, we have expanded the content of each caption to include detailed descriptions of the datasets, explanations of the abbreviations used, and summaries of the key results, ensuring that all figures and tables are self-explanatory in the revised manuscript.]
Comments 13:[ Minor formatting issues in references: Some references (e.g., Lines 568–575) include inconsistent punctuation or spacing around years, volumes, and page numbers. These should align with the journal’s citation style.]
Response 13:[ Thank you for your reminder, we have made revisions accordingly.]
Round 2
Reviewer 2 Report
Comments and Suggestions for Authors
I thank the authors for responding to my comments. The revised manuscript has been improved. I have no further suggestions.
Author Response
We are grateful for your positive feedback on the revised manuscript and your efforts in evaluating our work. We sincerely appreciate your approval of the revisions we made.
Reviewer 3 Report
Comments and Suggestions for Authors
The revision has substantially improved the manuscript’s clarity and presentation. I recommend acceptance after minor revision. The remaining issues are limited to minor editorial cleanup, mainly eliminating small instances of duplication and standardizing spacing and punctuation. For example, the text still contains a duplicated word (“4-bp deletion deletion”).” In addition, a few places show missing spaces after commas and inconsistent punctuation/spacing around statistical terms which should be standardized throughout.
Author Response
comments1:[The revision has substantially improved the manuscript’s clarity and presentation. I recommend acceptance after minor revision. The remaining issues are limited to minor editorial cleanup, mainly eliminating small instances of duplication and standardizing spacing and punctuation. For example, the text still contains a duplicated word (“4-bp deletion deletion”).” In addition, a few places show missing spaces after commas and inconsistent punctuation/spacing around statistical terms which should be standardized throughout.]
Reply:[Thank you sincerely for your positive evaluation and constructive suggestions on our manuscript. We greatly appreciate your recognition of the improved clarity and presentation after revision. We have carefully addressed all the minor editorial issues you pointed out: The duplicated word “deletion” in the phrase “4-bp deletion deletion” has been removed. All missing spaces after commas throughout the text have been supplemented. Punctuation and spacing around statistical terms have been standardized uniformly in accordance with the journal’s guidelines.]